# High-Precision Inertial Sensor Charge Ground Measurement Method Based on Phase-Sensitive Demodulation

**DOI:** 10.3390/s24031009

**Published:** 2024-02-04

**Authors:** Yang Liu, Tao Yu, Yuhua Wang, Zihan Zhao, Zhi Wang

**Affiliations:** 1Changchun Institute of Optics, Fine Mechanics and Physics, Chinese Academy of Sciences, Changchun 130033, China; liuyang21h@mails.ucas.ac.cn (Y.L.); wangyuhua21@mails.ucas.ac.cn (Y.W.); zhaozihan22@mails.ucas.ac.cn (Z.Z.); 2University of Chinese Academy of Sciences, Beijing 100049, China; 3School of Fundamental Physics and Mathematical Sciences, Hangzhou Institute for Advanced Study, University of Chinese Academy of Sciences, Hangzhou 310024, China

**Keywords:** space gravitational wave detection, inertial sensor, charge measurement, force modulation method, phase-sensitive demodulation

## Abstract

Inertial sensors are the key payloads in space gravitational wave detection missions, and they need to ensure that the test mass (TM), which serves as the inertial reference, freely floats in the spacecraft without contact, so that the TM is not disturbed by the satellite platform and the cosmic environment. Space gravitational wave detection missions require that the residual acceleration of the TM should be less than $3\times 10^{-15}\;{\mathrm{ms}}^{-2}{\mathrm{Hz}}^{-1/2}$. However, the TM with charges will interact with surrounding conductors and magnetic fields, introducing acceleration noise such as electrostatic force and Lorentz force. Therefore, it is necessary to carry out charge management on the TM, in which the high-precision measurement of charge is crucial. Space gravitational wave detection missions require a residual charge measurement accuracy of $3\times 10^{-13}\;C$ for the TM. In this paper, we design a high-precision inertial sensor charge measurement method based on phase-sensitive demodulation (PSD). By establishing a torsion pendulum rotation model based on the force modulation method, the characteristics of the TM torsion angle signal are analyzed. The PSD is used to extract the amplitude of the specific frequency signal component containing the charge information, and then to calculate the value of the accumulated charges. The method is compared with the Butterworth band-pass filtering method, and the simulation results show that the method has a higher measurement accuracy, shorter settling time, and stronger anti-interference ability, meeting the TM residual charge measurement accuracy index requirement.

## 1. Introduction

Inertial sensors are the key payloads in space gravitational wave detection missions, and their cores are a sensitive structure consisting of the TM and the electrode housing (EH) [1]. The TM serves as the spacecraft’s inertial reference, and the inertial sensors need to ensure that the TM is freely suspended within the spacecraft without contact [2]. Although the spacecraft is able to block most of the external environmental disturbances, some cosmic rays and energetic particles can penetrate the spacecraft and directly interact with the TM, resulting in the accumulation of charges on the TM.

The charges accumulated on the TM interact with the surrounding conductors to generate electrostatic forces. In addition, the accumulated charges also couple with the interplanetary magnetic fields and spacecraft internal magnetic fields to generate the Lorentz force. These all result in additional acceleration noises [3]. Therefore, charge management is necessary for the TM. With charge management, the residual charges on the TM are required to be less than $2\times 10^{7}e$ (where *e* represents an electron, $1e=1.6\times 10^{-19}\;C$ and $2\times 10^{7}e=3.2\times 10^{-12}\;C$) for LISA (Laser Interferometer Space Antenna) [4], and less than $10^{6}e$ ($1.6\times 10^{-13}\;C$) for Tianqin [5,6], so the residual charge measurement accuracy of $3\times 10^{-13}\;C$ is required for the TM.

The charge accumulation on the TM is mainly caused by galactic cosmic rays (GCRs) and solar high-energy particles (SEPs). According to the existing studies on the rate of charge accumulation on the TM, the rate of charge accumulation on the TM due to GCRs is in the range of $+10es^{-1}$ to $+100es^{-1}$ [7,8,9,10,11,12,13,14,15], assuming $+100es^{-1}$ as the maximum rate for the calculation, the TM can be charged to a level of $6\times 10^{7}e$ ($9.6\times 10^{-12}\;C$) in about one week. On the other hand, a large SEP event that can reach a maximum charging rate of about +70,000$es^{-1}$ may only occur once in a solar cycle, with a total charge deposition of $+2\times 10^{9}e$ ($3.2\times 10^{-10}\;C$) during a five-day SEP event [13]. Therefore, for short-term charge accumulation in the space environment, the total amount of charge can be focused on less than or equal to $10^{-10}\;C$.

The existing research on charge measurement and estimation methods mainly include the force modulation charge measurement method, which has been verified to be feasible by the GP-B (Gravity Probe B), LISA Pathfinder, etc. [16,17], as well as the disturbance-free charge estimation method proposed by the Tianqin team [18]. The force modulation method is a commonly used measurement method in space missions, which is based on the principle of applying signals to the electrodes around the TM along the sensitive axis to stimulate the motion of the TM, and estimating the charge through its position and attitude changes. However, charge measurement using the force modulation method carries the risk of interrupting gravitational wave detection. To address this problem, the Tianqin team proposed a disturbance-free charge estimation method. By analyzing the motion of the TM, the accumulated charge information embedded in its motion signal can be solved. And the method needs to extract specific frequency terms containing charge information from multiple AC components, but due to the similarity of the frequencies of the AC components, it is difficult to accurately extract specific frequency components through a simple filter, which will affect the accuracy of the results. The effectiveness of this method has only been verified through simulations.

To fulfill the ground testing requirements of inertial sensors, this paper integrates the characteristics of the ground testing device, a torsion pendulum, and establishes the torsion pendulum rotation model using the force modulation method. On this basis, a thorough analysis is conducted on the characteristics of the torsion angle signal. This paper has designed a charge measurement method based on PSD, which can resolve the charges accumulated on the TM with a higher accuracy. PSD uses reference signals of the same frequency as the excitation signal to demodulate the amplitude and phase information of the measured signal, which has the advantage of effectively suppressing the interference or noise that is not of the same frequency as the excitation frequency [19]. The amplitude of the specific frequency component containing the charge information can be extracted more accurately by PSD, resulting in a higher accuracy and meeting the ground measurement needs of the charge management system on inertial sensors.

## 2. Rotational Model of the Torsion Pendulum Based on the Force Modulation Method

In space, the TM inside the inertial sensor is in free suspension, while on the ground due to the influence of Earth’s gravity, such a free state is difficult to realize, so the ground test of the charge measurement system is a very difficult task. A torsion pendulum can simulate the free suspension state of the TM in space due to its special structure, which has a free rotational degree of freedom around the z-axis. Moreover, the torsion pendulum is sensitive to the tiny torques in the horizontal direction, and it can measure very small forces with high accuracy [20]. Therefore, the torsion pendulum can be used to test and evaluate the key technologies and performance of inertial sensors on the ground.

### 2.1. TM Torque Driver Model

The TM is suspended by a suspension fiber to form the most basic torsion pendulum system. The TM consists of a hollow gold-coated Al cube, which is surrounded by eighteen electrodes, including twelve sensing/actuation electrodes (grey) and six injection electrodes (yellow) distributed as shown in Figure 1, with two sensing/actuation electrodes on each face, one injection electrode on each y face and two injection electrodes on each z face [21]. During operation, the position and attitude of the TM inside the EH are sensed and controlled by the six-degrees-of-freedom differential capacitor pairs consisting of twelve sensing/actuation electrodes [22,23].

The TM of mass m and the electrodes on the EH form parallel capacitors that control the TM by applying the voltages to the sensing electrodes. The electrostatic force between the TM and the electrodes can be described as follows: (1)$$F=\frac{1}{2}\frac{\partial C_{i}}{\partial q}{\left(u_{i}-u_{TM}\right)}^{2}$$
where $C_{i}$ is the capacitance formed by the TM and the ith electrode, *q* is a general coordinate, $u_{i}$ is the voltage on the ith electrode, and $u_{TM}$ is the surface potential of the TM, which can be expressed as follows: (2)$$u_{TM}=\frac{Q_{TM}}{C_{T}}+\frac{1}{C_{T}}\sum_{i}C_{i}u_{i}$$
where $Q_{TM}$ is the charges accumulated on the TM and $C_{T}$ is the total capacitance inside the EH.

Let the voltages applied on the four electrodes on the sensitive x axis be $u_{i}$, i=1,…,4, and the electrostatic force between TM and electrode 1 can be expressed as follows: (3)$$F_{1}=\frac{\varepsilon S}{2d_{0}^{2}}{\left(u_{1}-u_{TM}\right)}^{2}$$
where ε is the vacuum dielectric constant, $\varepsilon =8.85\times 10^{-12}\;F/m$, *S* is the effective area of the capacitor consisting of the TM and the electrodes, and $d_{0}$ is the distance between the TM and the electrodes when the TM is at the center of the EH.

The forces between the TM and the three remaining electrodes can also be obtained by the same method. By applying specific polarity voltages to the four electrodes surrounding the TM along the x-axis direction, the TM can rotate in φ degree of freedom, as shown in Figure 2.

For the TM rotation, the voltages applied to each electrode in the x axis are as follows: (4)$$\left\{\begin{array}{c} u_{1\varphi }={+u}_{TS,1\varphi }\sin\left(2\pi f_{\varphi }t\right)\\ u_{2\varphi }={-u}_{TS,1\varphi }\sin\left(2\pi f_{\varphi }t\right)\\ u_{3\varphi }={+u}_{TS,1\varphi }\sin\left(2\pi f_{\varphi }t\right)\\ u_{4\varphi }={-u}_{TS,1\varphi }\sin\left(2\pi f_{\varphi }t\right) \end{array}\right.$$

Assuming that the TM is at the center of the EH, there are the same capacitances formed by the TM and the four electrodes in the x-axis direction, so,
(5)$$u_{TM}=\frac{Q_{TM}}{C_{T}}+\frac{1}{C_{T}}\sum_{i=1}^{4}C_{i}u_{i}=\frac{Q_{TM}}{C_{T}}$$

The TM is subjected to a combined electrostatic torque of
(6)$$\begin{array}{c} M_{\varphi }=\frac{b}{2}\left[\left(F_{1}-F_{3}\right)-\left(F_{2}-F_{4}\right)\right]=\frac{2\varepsilon bS}{d_{0}^{2}}u_{TM}u_{TS,1\varphi }\sin\left(2\pi f_{\varphi }t\right) \end{array}$$
where *b* is the distance between the centers of the two electrodes on the same side, $A=\frac{2\varepsilon bS}{d_{0}^{2}}u_{TM}u_{TS,1\varphi }$.

### 2.2. Dynamic Model and Frequency Response Analysis of the Torsion Pendulum

Neglecting the noise interference, the torsion pendulum is subjected to three effects:The torque τ(t) of interaction between the charged TM and the surrounding conductors;The damping torque of vibration;The elastic restoring torque.

Usually, the torsion pendulum system works in a high vacuum environment, the air damping is much smaller than the structural damping of the system, and the dynamics model of the torsion pendulum can be reduced to a second-order mass-spring-damper model, the dynamics equation of which is
(7)$$I_{r}\ddot{\varphi }\left(t\right)+\gamma \dot{\varphi }\left(t\right)+\Gamma \varphi \left(t\right)=\tau \left(t\right)$$
where $I_{r}$ is the rotational inertia of the torsion pendulum, γ is the damping coefficient, $\gamma =\Gamma /(\omega_{n}Q)$, *Q* is the quality factor of the torsion pendulum, Γ is the restoring stiffness of the suspension fiber, $\omega_{n}=\sqrt{\left(\Gamma /I_{r})\right)}$ is the intrinsic frequency of the torsion pendulum, φ is the torsion angle, and τ is the torque.

The Laplace transform of Equation (Equation 7) gives
(8)$$I_{r}s^{2}\varphi \left(s\right)+\gamma s\varphi \left(s\right)+\Gamma \varphi \left(s\right)=\tau \left(s\right)$$
where φ(s) and τ(s) are the Laplace transforms of the torsion angle and torque, respectively, and s=jω is the Laplace variable.

Then the transfer function $H_{\tau ,\varphi }\left(s\right)$ between the torque and torsion angle is
(9)$$H_{\tau ,\varphi }\left(s\right)=\frac{\varphi \left(s\right)}{\tau \left(s\right)}\;=\frac{1}{I_{r}s^{2}+\gamma s+\Gamma }$$

The standard form of the second-order system is $G\left(s\right)=\frac{\omega_{n}^{2}}{s^{2}+2\xi \omega_{n}s+\omega_{n}^{2}}$, where $\xi =\gamma /(2\sqrt{\left(I_{r}\Gamma \right)})$ is the damping ratio. Equation (Equation 9) can be further transformed into the following form: (10)$$H_{\tau ,\varphi }\left(s\right)=\frac{1}{\Gamma }\cdot \frac{\omega_{n}^{2}}{s^{2}+2\xi \omega_{n}s+\omega_{n}^{2}}$$

Let s=jω in Equation (Equation 10), then,
(11)$$H_{\tau ,\varphi }\left(j\omega \right)=\frac{1}{\Gamma }\cdot \frac{1}{{-\left(\frac{\omega }{\omega_{n}}\right)}^{2}+2\xi \frac{\omega }{\omega_{n}}j+1}$$

Then, let $\Omega =\omega /\omega_{n}$, which yields
(12)$$\begin{array}{cc} H_{\tau ,\varphi }\left(j\omega \right)\; & =\frac{1}{\Gamma }\cdot \left(\frac{{1-\Omega }^{2}}{{\left(1-\Omega^{2}\right)}^{2}+4\xi^{2}\Omega^{2}}-\frac{2\xi \Omega }{{\left(1-\Omega^{2}\right)}^{2}+4\xi^{2}\Omega^{2}}j\right) \end{array}$$

So, the amplitude response of the torsion pendulum is
(13)$$\left|H_{\tau ,\varphi }\left(j\omega \right)\right|=\frac{1}{\Gamma }\cdot \sqrt{\frac{1}{{\left(1-\Omega^{2}\right)}^{2}+4\xi^{2}\Omega^{2}}}$$
where its amplitude response is $∠H_{\tau ,\varphi }\left(j\omega \right)$.

### 2.3. Rotational Model of the Torsion Pendulum Based on the Force Modulation Method

The sinusoidal voltages of certain amplitude and frequency are applied to the four sensing electrodes around the TM along the x axis, and due to the existence of the potential difference between the electrodes and the TM, electrostatic interactions will occur, the TM produces a torsional motion, and the torsion angle of the TM can be measured. On the basis of the above mathematical derivation, a model is built in Matlab/Simulink, as shown in Figure 3.

The inputs of the model are the initial set charge value $Q_{1}$, as well as the amplitude $u_{TS,1\varphi }$ and frequency $f_{\varphi }$ of the excitation applied to the electrodes, and the output is the torsion angle of the TM, and the parameters in the model are shown in Table 1. Based on the simulation of the model and the analysis of the angle signal both in the time domain and frequency domain, it can be found that the angle signal mainly consists of two frequency components, which are the frequency $f_{\varphi }$ of the applied excitation and the intrinsic frequency $f_{n}$ of the torsion pendulum.

According to Equations (Equation 6) and (Equation 13), if applying the excitation signal with the amplitude of $u_{TS,1\varphi }$ and the frequency of $f_{\varphi }$ to each electrode on the x axis, we can obtain the output angle signal ideally as
(14)$$\varphi \left(t\right)=A_{1}\sin\left(2\pi f_{\varphi }t+\varphi_{1}\right)+A_{2}\sin\left(2\pi f_{n}t+\varphi_{2}\right)$$
where the signal component with frequency $f_{\varphi }$ has amplitude $A_{1}$, and the signal component with frequency $f_{n}$ has amplitude $A_{2}$, with $A_{1}=A\cdot \left|H_{\tau ,\varphi }\left(j\omega \right)\right|=\frac{2\varepsilon bS}{d_{0}^{2}}u_{TM}u_{TS,1\varphi }\left|H_{\tau ,\varphi }\left(j\omega \right)\right|$.

Also, since $Q_{TM}=u_{TM}C_{T}$, the information about the accumulated charges on the TM is embedded in the amplitude of the signal component with frequency $f_{\varphi }$ in the output angle signal. Therefore, it is necessary to consider extracting the amplitude of the signal component with frequency $f_{\varphi }$ accurately by analyzing and processing the output angle signal.

The realization of the torsion pendulum is to maximize its sensitivity in the millihertz frequency range of interest for space gravitational wave detection, so that its resonance (the maximum angle response of the pendulum to a given torque) is in the millihertz frequency range [24]. In order to obtain a higher sensitivity of the torsion pendulum, the frequency of the applied excitation also needs to be chosen in the millihertz range. Taking the sinusoidal excitation with amplitude $u_{TS,1\varphi }=3\;V$ and frequency $f_{\varphi }=0.003\;\mathrm{Hz}$ applied to the model as an example, the resulting angle waveform is shown in Figure 4.

Through performing a frequency domain analysis, as shown in Figure 5, the superposition of two sine waves with frequencies of $f_{n}=0.00177\;\mathrm{Hz}$ and $f_{\varphi }=0.003\;\mathrm{Hz}$ can be observed.

Due to the small difference between the frequencies, it is difficult to accurately extract the amplitude of the signal component the frequency of which is $f_{\varphi }$ through a simple filter. Therefore, the PSD method is selected.

## 3. Charge Calculation Method Based on PSD

### 3.1. Design of Charge Calculation Method Based on PSD

PSD is based on the principle of signal cross-correlation, which determines the correlation between two signals by calculating their cross-correlation function. When the frequencies of the two signals are the same, the cross-correlation function is the periodic components of the double frequency. When the frequencies of the two signals are different, the cross-correlation function is zero, indicating that the two signals are uncorrelated. The principle diagram is shown in Figure 6.

In Figure 6, $V_{in}$ is the input signal, $r_{sin}$ and $r_{cos}$ are the sine and cosine reference signals, $d_{r}$, $d_{q}$ are the in-phase and quadrature components obtained by demodulation, and *A* and φ are the amplitude and phase of the signal components at the desired specific frequency, respectively. The analog multiplier demodulation uses analog multipliers and low-pass filters to perform cross-correlation operations on the signals, while the digital multiplier demodulation method uses multiplication accumulators [25,26].

PSD is performed with the ideal torsion angle signal as the input signal, and the sine and cosine reference signals are introduced as $r_{1}\left(t\right)=\sin\left(2\pi f_{\varphi }t-\phi \right)$, $r_{2}\left(t\right)=\cos\left(2\pi f_{\varphi }t-\phi \right)$, so the results of the cross-correlation operations on the torsion angle signal and the sine and cosine reference signals are as follows: (15)$$\begin{array}{cc} d_{r}\left(t\right)\; & =\varphi \left(t\right)\cdot r_{1}\left(t\right)\\ \; & =\frac{A_{1}}{2}\cos\phi -\frac{A_{1}}{2}\cos\phi \cos\left(4\pi f_{\varphi }t\right)-\frac{A_{1}}{2}\sin\phi \sin\left(4\pi f_{\varphi }t\right)+\\ \; & A_{2}\cos\phi \sin\left(2\pi f_{\varphi }t\right)\sin\left(2\pi f_{n}t\right)-A_{2}\sin\phi \sin\left(2\pi f_{n}t\right)\cos\left(2\pi f_{\varphi }t\right) \end{array}$$
and
(16)$$\begin{array}{cc} d_{q}\left(t\right)\; & =\varphi \left(t\right)\cdot r_{2}\left(t\right)\\ \; & =\frac{A_{1}}{2}\sin\phi +\frac{A_{1}}{2}\cos\phi \sin\left(4\pi f_{\varphi }t\right)-\frac{A_{1}}{2}\sin\phi \cos\left(4\pi f_{\varphi }t\right)+\\ \; & A_{2}\cos\phi \sin\left(2\pi f_{n}t\right)\cos\left(2\pi f_{\varphi }t\right)+A_{2}\sin\phi \sin\left(2\pi f_{n}t\right)\sin\left(2\pi f_{\varphi }t\right) \end{array}$$

From Equations (Equation 15) and (Equation 16), it can be found that the cross-correlation operation results contain a DC component and AC components, while the DC component amplitude contains the charge information. And the DC component amplitude can be obtained by filtering out the AC components using a low-pass filter, which can then be used to calculate the value of the charges.

After filtering out the AC components using a low-pass filter, we can obtain
(17)$$\left\{\begin{array}{c} d_{r}=\frac{A_{1}}{2}\cos\phi \\ d_{q}=\frac{A_{1}}{2}\sin\phi \end{array}\right.$$
where $d_{r}$ is called the in-phase component and $d_{q}$ is called the quadrature component, from which the amplitude and phase of the signal component at the desired specific frequency can be calculated.
(18)$$\left\{\begin{array}{c} \hat{A_{1}}=2\sqrt{d_{r}^{2}+d_{q}^{2}}\\ \hat{\phi }=arctan\frac{d_{q}}{d_{r}} \end{array}\right.$$

The charge $Q_{TM}$ can also be solved from $A_{1}=\frac{2\varepsilon bS}{d_{0}^{2}}u_{TM}u_{TS,1\varphi }\left|H_{\tau ,\varphi }\left(j\omega \right)\right|$ and $Q_{TM}=u_{TM}C_{T}$: (19)$$Q=\frac{d_{0}^{2}C_{T}\sqrt{d_{r}^{2}+d_{q}^{2}}}{\varepsilon bSu_{TS,1\varphi }\left|H_{\tau ,\varphi }\left(j\omega \right)\right|}$$

If the phase of the output angle signal is not zero, similarly, it can be deduced that the results of the cross-correlation contain DC components and AC components, and the amplitudes of the DC components can be obtained through a low-pass filter. However, the DC amplitudes also contain the sine and cosine values of the phase of the angle signal, so the phase of the output angle signal needs to be detected first.

### 3.2. Model of Charge Measurement in the Torsion Pendulum Based on PSD

The following simulation model is established in MATLAB/Simulink, as shown in Figure 7. The inputs of the model are the initial set charge value $Q_{1}$, the amplitude $u_{TS,1\varphi }$, and frequency $f_{\varphi }$ of the excitation applied to the electrodes, and the output is the estimated charge value $Q_{e}$. The TM is twisted by applying excitation to the four electrodes around the TM along the sensitive axis, and the resulting torque is converted into an angle signal by the transfer function of the torsion pendulum, which is then subjected to PSD to calculate the value of the charges.

The initial charge setting value $Q_{1}$ is taken as the agreed true value and the simulation value $Q_{e}$ is taken as the measurement value, so the absolute error is $\Delta Q=Q_{e}-Q_{1}$ and the relative error is $\delta =\frac{Q_{e}\;-\;Q_{1}}{Q_{1}}$.

### 3.3. Performance Verification

In order to verify the effectiveness and anti-interference performance of the designed charge measurement method based on PSD, we compared the method with three common band-pass filtering methods, namely the Butterworth BPF, the Chebyshev II BPF, and the Bessel BPF in the ideal case and with noise introduced, respectively. We chose two representative charge values as the initial set charge values, $1\times 10^{-12}\;C$ and $1\times 10^{-13}\;C$. The simulation results of the different methods in the ideal case and with noise introduced are summarized in Table 2, respectively.

#### 3.3.1. In the Ideal Case

Figure 8 shows the simulation results of these four methods in the ideal case, where the black curve represents the simulation result obtained using the designed charge measurement method based on PSD, the red curve represents Butterworth BPF, the yellow curve represents ChebyshevII BPF and the purple curve represents Bessel BPF, respectively. The results show that in the ideal case, regardless of the initial charge setting value $Q_{1}=1\times 10^{-12}\;C$ or $Q_{1}=1\times 10^{-13}\;C$, the estimation error of PSD is smaller than that of the other three methods. The time required for the charge simulation results to reach the initial charge setting value and remain within ±2% error is defined as the settling time (Ts), the Ts required by PSD is also shorter than that required by Butterworth BPF or ChebyshevII BPF. Therefore, from a comprehensive perspective, PSD has good effects in both Ts and accuracy.

By giving multiple sets of initial set charge values $Q_{1}\in \left[1,1\times 10^{-25}\right]\;C$, multiple sets of simulation results are obtained, and the absolute and relative errors under different initial set charge values are plotted as shown in Figure 9a,b, respectively. The results show that both the absolute and relative errors of the PSD results are significantly better than those of the other three methods results across the full range.

The root mean square error (RMSE) is the square root of the ratio of the sum of the squares of the deviations of the measured value from the true value to the number of observations *m*. It is used to measure the deviation between the measured value and the true value. In addition, the mean absolute error (MAE) is the average absolute value of the error, and the mean absolute percentage error (MAPE) is the average relative error. The smaller the RMSE, MAE and MAPE, the higher the measurement accuracy. Their formulas are as follows: (20)$$RMSE=\sqrt{\frac{\sum_{i=1}^{m}{\left(x\left(i\right)-y\left(i\right)\right)}^{2}}{m}}$$
(21)$$MAE=\frac{1}{m}\sum_{i=1}^{n}\left|x\left(i\right)-y\left(i\right)\right|$$
(22)$$MAPE=\frac{100\%}{m}\sum_{i=1}^{n}|\frac{x\left(i\right)-y\left(i\right)}{y(i)}|$$
where x(i) is the ith measurement value, y(i) is the ith truth value, and *m* is the number of observations.

For the short-term charge accumulation in the space environment, the total amount of charges that is mainly concerned in the charge measurement should be less than or equal to $1\times 10^{-10}\;C$. In the ideal case, both the MAE and MAPE of the method based on PSD are less than those of the other three methods, as shown in Table 2. Most importantly, the RMSE obtained by the PSD method is $1.256\times 10^{-13}\;C$, which is smaller than the TM residual charge measurement accuracy index of $3\times 10^{-13}\;C$. This shows that the designed charge measurement method based on PSD meets the TM charge measurement accuracy index requirement.

#### 3.3.2. With Charge Noise and Angle Sensor Noise Introduced

The charge accumulation on the TM is caused by two main types of particle radiation, GCRs and SEPs, which are driven by certain types of solar events. Although the charging rate caused by GCR events is relatively stable, it also varies with the solar activity cycle. The GCR flux reaches its maximum during the solar minimum and its minimum during the solar maximum. On the other hand, SEP events are short lived and originate from solar eruptions. SEPs vary with solar activity, lasting from a few hours to a few days, but can temporarily increase the cosmic ray flux by several orders of magnitude. SEP events are expected to occur only a few times per year during the solar maximum, and less than once per year during the solar minimum. In addition, due to the complexity of the cosmic environment, there are still unknown sources of charge accumulation in space other than GCRs and SEPs, which can cause variations in the charge accumulation on the TM. Considering these factors that cause charge fluctuations, charge noise is introduced into the model of charge measurement in the torsion pendulum based on PSD. Assuming that the initial set charge value carries a zero-mean Gaussian white noise with variance $\sigma =10^{-13}\;C$, the noise curve is shown in Figure 10.

In addition, in the TM torsion angle measurement step, noise may be introduced due to the limitations of measurement methods and measurement devices, so the angle sensor noise is introduced in the the model, and the angle signal output from the torsion pendulum rotation model based on the force modulation method is combined with a zero-mean, Gaussian white noise with a variance of $\sigma =10^{-5}\;\mathrm{rad}$ as the real angle signal. The noise curve is shown in Figure 11.

With the introduction of charge noise and angle sensor noise, the designed charge measurement method based on PSD is compared with the normal filtering methods using Butterworth BPF, ChebyshevII BPF, and Bessel BPF, similarly. Figure 12 shows the simulation results of these four methods with noise introduced. The black curve represents the simulation result obtained using PSD and the red curve represents Butterworth BPF, the yellow curve represents ChebyshevII BPF, and the purple curve represents Bessel BPF, respectively. The simulation results are shown in Table 2. The results show that when noise is introduced, regardless of the initial charge setting value $Q_{1}=1\times 10^{-12}\;C$ or $Q_{1}=1\times 10^{-13}\;C$, the estimation error of the PSD is significantly smaller than that of the other three methods, with stronger anti-interference capability and shorter Ts required.

By giving multiple sets of initial set charge values $Q_{1}\in \left[1,1\times 10^{-15}\right]\;C$, multiple sets of simulation results are obtained, and the absolute and relative errors under different initial set charge values are plotted as shown in Figure 13a,b, respectively. The results show that the results of PSD are significantly better than those of the other three methods in terms of both the absolute and relative errors.

Under the consideration of the charge noise and angle sensor noise, for the charge measurement, the main concern of the charge magnitude, i.e. less than or equal to $1\times 10^{-10}\;C$, both the MAE and MAPE of the method based on PSD are less than those of the other three methods, as shown in Table 2. Most importantly, the RMSE obtained by PSD is $2.051\times 10^{-13}\;C$, which is less than the TM residual charge measurement accuracy index of $3\times 10^{-13}\;C$. This shows that the designed charge measurement method based on PSD meets the TM charge measurement accuracy index requirement.

## 4. Conclusions

The accurate measurement of the charges accumulated on the TM is of great significance. Firstly, it is conducive to the analysis of the various noises caused by the accumulated charges on the TM, which provides a reference for the construction of high-precision inertial sensors. Secondly, it provides a basis for the subsequent charge control. By accurately measuring the charges accumulated on the TM, we can determine the control mode that the TM charge management system should adopt. This paper studies the ground measurement method of inertial sensor charges and completes the establishment and simulation analysis of the torsion pendulum rotation model based on the force modulation method. By analyzing the characteristics of the TM torsion angle signal, a high-precision inertial sensor charge measurement method based on PSD is designed. The results show that when in the ideal case and considering the charge noise and angle sensor noise, the method meets the TM charge measurement accuracy index requirement, with a high accuracy and strong anti-interference ability.

## Figures and Tables

**Figure 1 sensors-24-01009-f001:**
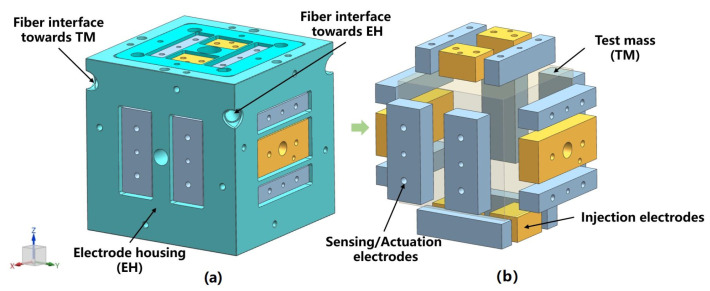
(**a**) Sensitive structure (including TM and EH), (**b**) 18 electrodes around the TM.

**Figure 2 sensors-24-01009-f002:**
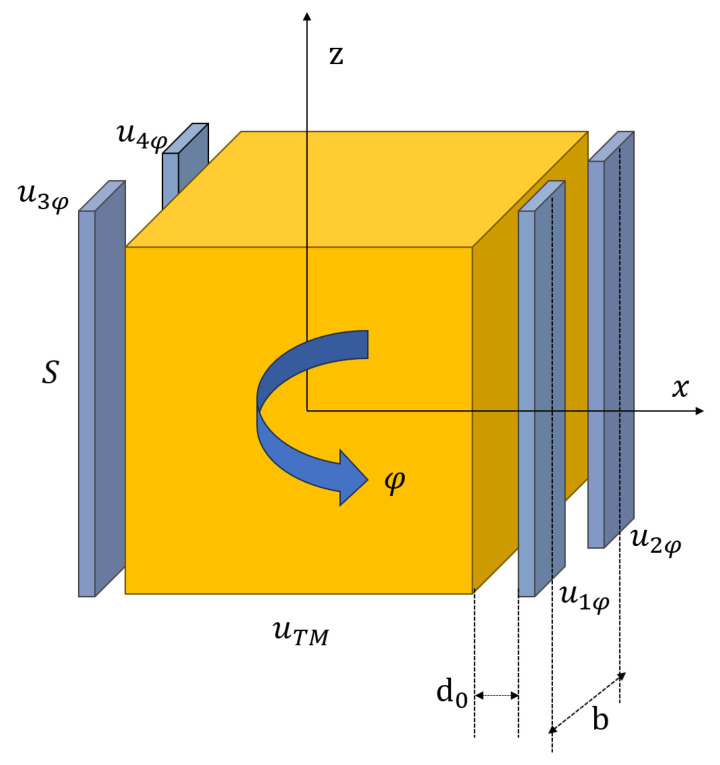
Actuation of TM in φ degree of freedom.

**Figure 3 sensors-24-01009-f003:**
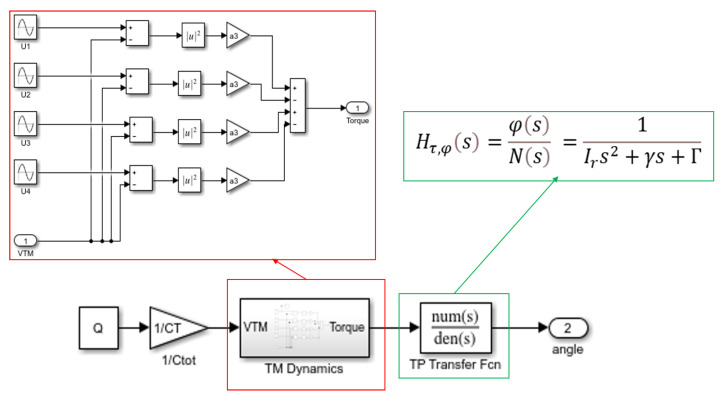
Rotational model of the torsion pendulum based on force modulation method.

**Figure 4 sensors-24-01009-f004:**
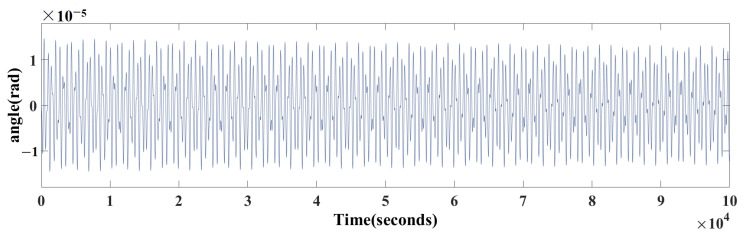
Time domain waveform of angle signal.

**Figure 5 sensors-24-01009-f005:**
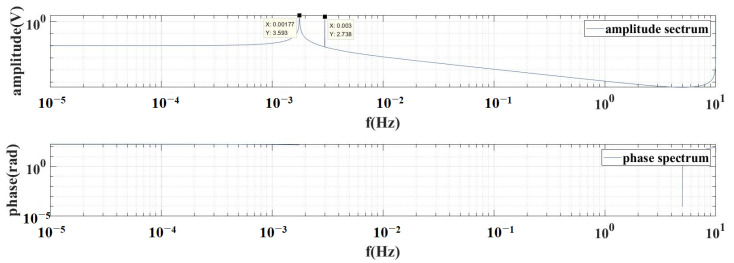
Amplitude spectrum and phase spectrum of angle signal.

**Figure 6 sensors-24-01009-f006:**
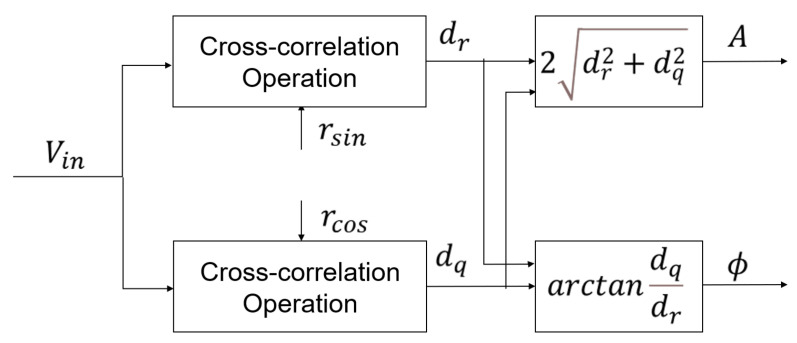
Principle diagram of PSD.

**Figure 7 sensors-24-01009-f007:**
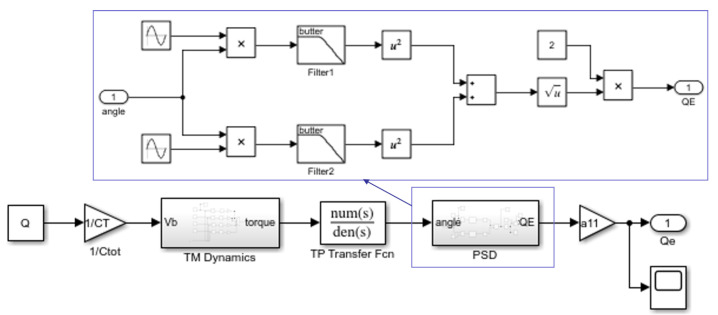
Charge measurement model of torsion pendulum based on PSD.

**Figure 8 sensors-24-01009-f008:**
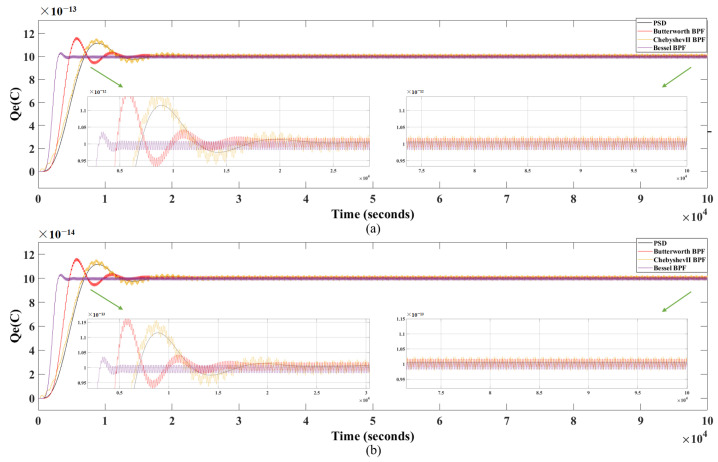
Simulation results of PSD, Butterworth BPF, Chebyshev II BPF and Bessel BPF methods for the ideal case (**a**) when $Q_{1}=1\times 10^{-12}\;C$ and (**b**) when $Q_{1}=1\times 10^{-13}\;C$.

**Figure 9 sensors-24-01009-f009:**
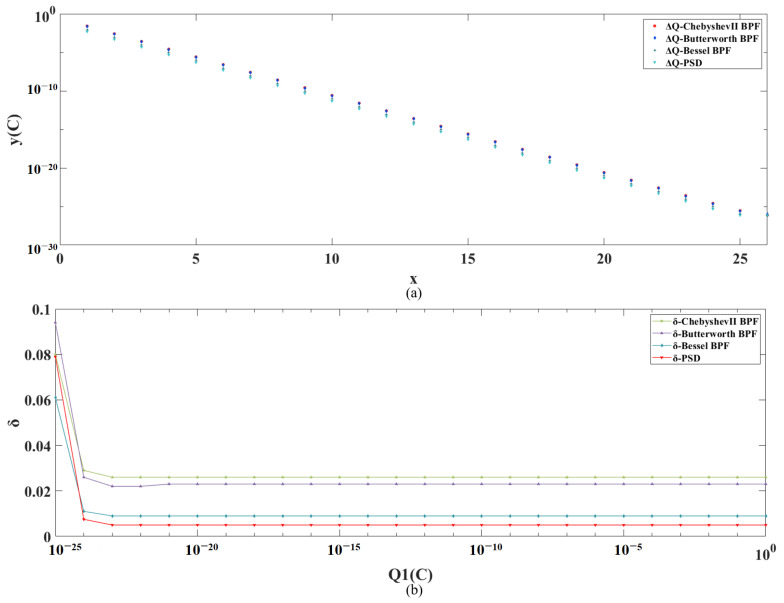
(**a**) Simulation results and corresponding absolute errors for multiple sets of simulations with PSD, Butterworth BPF, Chebyshev II BPF and Bessel BPF, in the ideal case, respectively. (**b**) Relative errors of PSD, Butterworth BPF, Chebyshev II BPF and Bessel BPF, for different initial set charge values in the ideal case, respectively.

**Figure 10 sensors-24-01009-f010:**
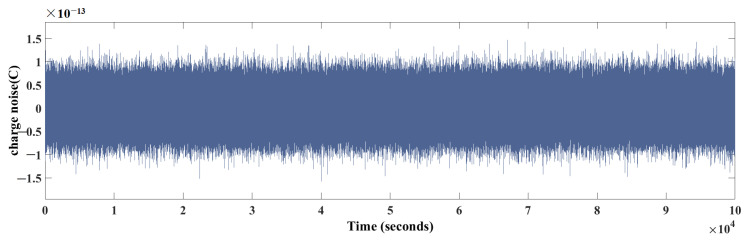
Charge noise.

**Figure 11 sensors-24-01009-f011:**
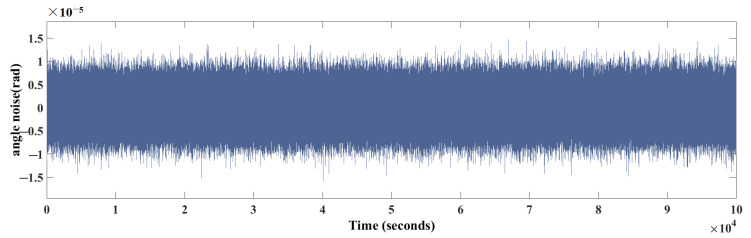
Angle sensor noise.

**Figure 12 sensors-24-01009-f012:**
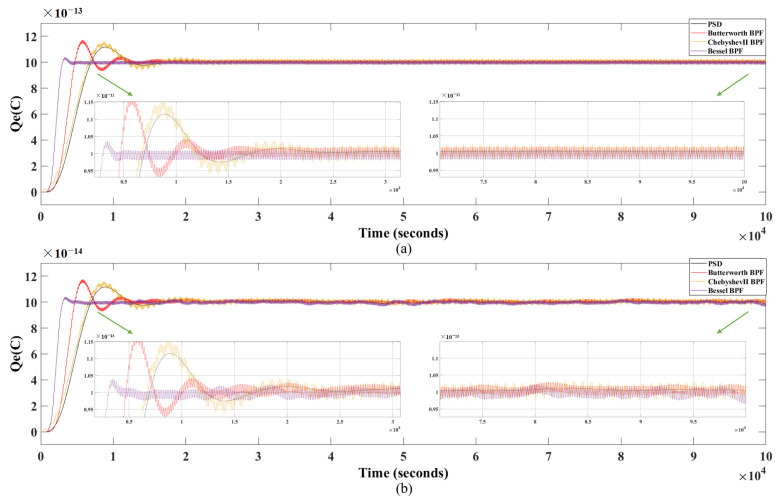
Simulation results of PSD, Butterworth BPF, Chebyshev II BPF and Bessel BPF methods with noise introduced (**a**) when $Q_{1}=1\times 10^{-12}\;C$ and (**b**) when $Q_{1}=1\times 10^{-13}\;C$.

**Figure 13 sensors-24-01009-f013:**
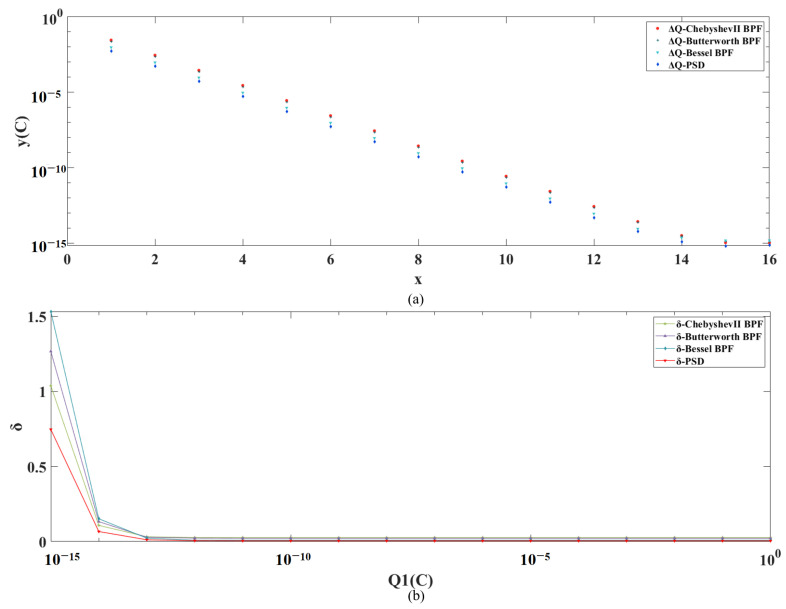
(**a**) Simulation results and corresponding absolute errors for multiple sets of simulations with PSD, Butterworth BPF, Chebyshev II BPF, and Bessel BPF with noise introduced, respectively. (**b**) Relative errors of PSD, Butterworth BPF, Chebyshev II BPF, and Bessel BPF for different initial set charge values with noise introduced, respectively.

**Table 1 sensors-24-01009-t001:** Parameters in the torsion pendulum rotation model based on the force modulation method.

Parameter	Meaning	Value
*m*	the mass of TM	1.96kg
$C_{T}$	the total capacitance inside the EH	$34.2\times 10^{-12}\;F$
ε	the vacuum dielectric constant	$8.85\times 10^{-12}\;F/m$
$d_{0}$	the distance between the TM and the electrodes when the TM is at the center of the EH	$4\times 10^{-3}\;m$
*b*	the distance between the centers of the two electrodes on the same side	$10.75\times 10^{-3}\;m$
$a_{3}$	factor	$7.7597\times 10^{-13}$
$I_{r}$	the rotational inertia of the torsion pendulum	$4.31\times 10^{-5}\;{\mathrm{kgm}}^{2}$
γ	the damping coefficient	$1.6557\times 10^{-10}$
*Q*	the quality factor of the torsion pendulum	2900
Γ	the restoring stiffness of the suspension fiber	$5.3491\times 10^{-9}\;\mathrm{Nm}/\mathrm{rad}$
$T_{0}$	free period of the torsion pendulum	564s
$\omega_{n}$	the intrinsic frequency of the torsion pendulum	$11.12\times 10^{-3}\;\mathrm{Hz}$
$u_{TS,1\varphi }$	the absolute value of amplitude of the excitation applied to the electrodes	3V
$f_{\varphi }$	the frequency of the excitation applied to the electrodes	$3\times 10^{-3}\;\mathrm{Hz}$
ξ	the damping ratio	$1.7241\times 10^{-4}$
Ω	frequency normalisation	1.6949

**Table 2 sensors-24-01009-t002:** Comparison of simulation results between PSD, Butterworth BPF, Chebyshev II BPF and Bessel BPF methods in the ideal case and with noise introduced.

		PSD	Butterworth BPF	ChebyshevII BPF	Bessel BPF
in the ideal case	RMSE	$1.256\times 10^{-13}\;C$	$4.522\times 10^{-13}\;C$	$6.533\times 10^{-13}\;C$	$2.261\times 10^{-13}\;C$
	MAE	$3.472\times 10^{-14}\;C$	$1.250\times 10^{-13}\;C$	$1.806\times 10^{-13}\;C$	$6.25\times 10^{-14}\;C$
	MAPE	0.00794	0.02015	0.02819	0.01107
with noise introduced	RMSE	$2.051\times 10^{-13}\;C$	$9.026\times 10^{-13}\;C$	$1.108\times 10^{-12}\;C$	$3.693\times 10^{-13}\;C$
	MAE	$9.310\times 10^{-14}\;C$	$4.081\times 10^{-13}\;C$	$5.004\times 10^{-13}\;C$	$1.674\times 10^{-13}\;C$
	MAPE	0.05575	0.10725	0.09556	0.11406

## Data Availability

The data and the source code are publicly available at https://github.com/YangLiucc/High-precision-Charge-Measurement-Method-Based-on-PSD.git (accessed on 24 November 2023).

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
