# Peer review of "High-Precision Inertial Sensor Charge Ground Measurement Method Based on Phase-Sensitive Demodulation"

_sensors, 2024, doi:10.3390/s24031009_

Round 1
Reviewer 1 Report
Comments and Suggestions for Authors
Why the work is limited to second order only.
The outcome is not quantified properly.
The methodology of the work needs more explanation.
How cross correlation is established and PSD.
Estimate the error of the system.
How many modes of TM in control operation , need to discuss
Check relevant good work titles.
Stabilization of Underactuated Mechanical System Using LQR Technique.
Validate/Verify the work properly
Reviewer 2 Report
Comments and Suggestions for Authors
Referee report for "High-precision Inertial Sensor Charge Ground Measurement Method Based on Phase Sensitive Demodulation"
This paper describes plans for constructing an instrument to study charge effect on test masses in the Earth laboratory. This is a necessary prerequisite before using such methods in space. The paper is generally well written, with only minor (and perfectly understandable) errors that I would expect to be fixed during the publication process.
I have several concerns about fundamental science that have caused a delay in writing this report.
1) the simulations in the paper assume white noise. But the instrument is designed for low-frequency measurements and the authors should expect to encounter 1/f noise from a variety of sources.
2) the text discusses having torsion balance resonance frequency in the millihertz range. This is a very low frequency and one would expect to see shoulders from the peak given a finite quality factor.
3) One would expect to see a degree of coupling of ground vibration to the torsion balance (btw, I think pendulum is reserved for apparatus where the restoring force is gravity). That coupling could be indirect such as by small changes in center of mass of of the balance. Unfortunately the frequency range of interest is sub-Hz and this is precisely where microseismic noise is large. Seismic isolation of some sort, together with underground operation might be desired.
4) The paper does not discuss thermal noise of the torsion balance. Because the torsion balance has non-zero damping it couples to the outside thermal bath and, in return, the thermal bath induces fluctuations in its position. As the authors opted to have the resonance frequency in millihertz, it would take an extraordinary quality factor to prevent coupling of thermal noise to the measurement region (where it will look as 1/f noise)
Now, considering the question of publication of the paper, the concerns 1-3) to a large degree are up to the authors - they might want to address them or, perhaps, leave the matter for future work.
I do feel strongly that the concern 4) - the thermal noise - needs to be addressed. At the very least if it was my paper I would not want to publish without making that calculation first. My experience with other devices is that millihertz resonant frequency coupled with reasonable quality factors and room temperature operation may result in levels of noise above authors requirement.
Theoretical paper discussing thermal noise (also recommend book by Braginsky et all "Systems with small dissipation")
Thermal noise in mechanical experiments
Peter R. Saulson
Phys. Rev. D 42, 2437 (1990)
Instrument paper briefly covering application to a balance - it has a formula to compute noise power density:
A high precision, compact electromechanical ground rotation sensor,
V. Dergachev, R. DeSalvo, M. Asadoor, A. Bhawal, P. Gong, C. Kim,
A. Lottarini, Y. Minenkov, C. Murphy, A. O'Toole, F. E. Peña Arellano,
A. V. Rodionov, M. Shaner, E. Sobacchi
Rev. Sci. Instrum. 85, 054502 (2014)
"... Moreover, the manuscript did not discuss in detail the potential issues that this method may have in the actual data processing, including its impact on the accuracy of charge measurement. If the team has an experimental platform, it is best for the authors to combine the experimental data to conduct the more realistic experimental verification. This will be more convincing. Overall, from the perspective of innovation and practicality, the current version of the manuscript is not suitable for publication. "
I think the referee misunderstands the purpose of the paper. In physics (at least) it is common to publish a paper describing design of the experiment before it begins construction. Very often the analysis (and simulations) are done by people with different skill sets from those that will get the experiment working. Publishing a paper is a way for them to get recognized.
Thus it is not reasonable to request that authors submit the paper after experimental data is collected - this might be years away. I also think some more specific comments are misdirected - but I would expect the paper authors to answer that.
The paper is generally well written, with only minor (and perfectly understandable) errors that I would expect to be fixed during the publication process.
Reviewer 3 Report
Comments and Suggestions for Authors
This paper introduced a charge measurement method based on PSD about the charges accumulated on TM, which is important in the construction of high-precision inertial sensors. A torsion pendulum rotation model based on the force modulation method is established, and the characteristics of the TM torsion angle signals are analyzed. The simulation model is established in MATLAB/Simulink. Assuming that the accumulated charge on TM is stable, a comparison was made between the Butterworth filter and the PSD method to verify the efficiency and accuracy of the PSD method. Then, with the introduction of charge noise and angle sensor noise and multiple sets of initial set charge values, the PSD and Butterworth BPF results show that PSD results are significantly better than those of Butterworth BPF in both absolute and relative errors. The PSD has stronger anti-interference capability and shorter the Ts.
However, there are some issues.
1. In equation (7)~(9), the variable about torque was inconsistent.
2. There was no explanation given as to why the sample response was not considered.
3. Is there a problem with the expression of equation (17)? Why is equation (17) the same as equation (15)~(16) after using a low-pass filter?
4. Suggest using multiple types of filters to compare with PSD.
5. The assumption of introducing charge noise and angle sensor noise is too simplistic.
Comments on the Quality of English LanguageThe language should be polished; thorough proofreading is suggested.
Reviewer 4 Report
Comments and Suggestions for Authors
This reviewer finds the paper to be high quality and worthy of publication. My concern is that the authors should have done noise not as a charge perturbation but as a perturbation moving the test mass (TM) off center. This means equation 5 is no longer true and the analysis for the accumulated charge is much more complicated. If the TM is off center, it is not clear that the phase sensitive demodulation can even work since applying voltages to the electrodes will cause lateral motion of the TM. Practically speaking, you need a robust measurement method that allows for tolerances.
Also, a reader is left wondering how accumulated charge is removed, but that is outside the authors' purpose.
Comments on the Quality of English LanguagePage 2, line 63 is a terrible run-on sentence. I would break it up into two or more sentences.
Reviewer 5 Report
Comments and Suggestions for Authors
This manuscript reports an inertial sensor charge measurement method based on phase sensitive demodulation (PSD). In fact, the PSD method has been widely used in the signal processing of torsion pendulum experimental data, such as the information extraction of the amplitude and frequency, and this manuscript just intended to apply this method to measure the inertial sensor charge. To verify the effectiveness of this method in charge measurement, the authors performed a numerical simulation, while it was too idealized and did not take into account specific experimental situations, such as the electric potential distribution, the attenuation and slow drift of the torsional pendulum motion, etc. Moreover, the manuscript did not discuss in detail the potential issues that this method may have in the actual data processing, including its impact on the accuracy of charge measurement. If the team has an experimental platform, it is best for the authors to combine the experimental data to conduct the more realistic experimental verification. This will be more convincing. Overall, from the perspective of innovation and practicality, the current version of the manuscript is not suitable for publication.
Round 2
Reviewer 1 Report
Comments and Suggestions for Authors
Most of the concerns are addressed. Manuscript can be accepted.
Author Response
Thank you again for your valuable comments.
Reviewer 2 Report
Comments and Suggestions for Authors
I think you misunderstand my comments on thermal noise. This is a fundamental noise that will always be present as long as your instrument components are at a certain temperature. You will not be able to isolate it by putting the instrument in a vacuum tank. The only ways to manage thermal noise is to either lower the temperature (i.e. go cryogenic) and/or use high-quality factor materials and design, which is challenging at low frequencies.
This is why making an estimate of thermal noise before construction is so important.
Marking "accept after minor revision" to give you an opportunity to make this calculation before the paper is published.
Comments on the Quality of English LanguageThe paper is well written and understandable.
Reviewer 5 Report
Comments and Suggestions for Authors
The comment is attached.
